# Metallography Specimen Mounting Device Suitable for Industrial or Educational Purposes

**Alfredo Márquez-Herrera** 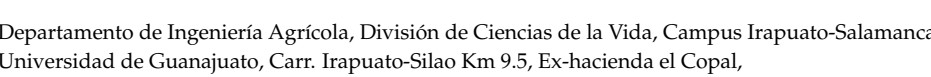

Departamento de Ingeniería Agrícola, División de Ciencias de la Vida, Campus Irapuato-Salamanca, Universidad de Guanajuato, Carr. Irapuato-Silao Km 9.5, Ex-hacienda el Copal, Irapuato 36500, Guanajuato, Mexico; amarquez@ugto.mx

**Abstract:** This work presents a novel, compact (six pieces), low-cost (<\$500 USD), and easy-to-manufacture metallography mounting device. The device is designed to produce high-quality polymer encapsulated samples that rival those obtained from commercial equipment (\$5000–\$10,000 USD). Utilizing the House of Quality (HoQ) framework within Quality Function Deployment (QFD), the device prioritizes critical customer requirements, including safety (validated via finite element method, FEM), affordability, and compatibility with standard hydraulic presses. FEM analysis under 29 MPa pressure revealed a maximum Von Mises stress of 80 MPa, well below the AISI 304 stainless steel yield strength of 170 MPa, yielding a static safety factor of 2.1. Fatigue analysis under cyclic loading (mean stress $\sigma_m$ = 40 MPa, amplitude stress $\sigma_a$ = 40 MPa) using the Modified Goodman Criterion demonstrated a fatigue safety factor of 3.75, ensuring infinite cycle durability. The device was validated at 140 °C (413.15 K) with a 5-min dwell time, encapsulating samples in a cylindrical configuration (31.75 mm diameter) using a 200 W heating band. Benchmarking confirmed performance parity with commercial systems in edge retention and surface uniformity, while reducing manufacturing complexity (vs. conventional 100-piece systems). This solution democratizes access to metallography, particularly in resource-constrained settings, fostering education and industrial innovation.

**Keywords:** metallography; device; polishing; QFD

## 1. Introduction

Metallography, a critical technique in materials science, involves the meticulous preparation and analysis of metallic and non-metallic samples to evaluate microstructural characteristics such as grain boundaries, phase distribution, surface hardness, and other characteristics [1,2]. These characteristics are vital for determining mechanical properties, corrosion resistance, thermal stability, etc., which are essential for applications in aerospace, automotive, and manufacturing industries [3,4]. The preparation process requires sequential conditioning steps: grinding with abrasive papers to remove surface irregularities, polishing with diamond or alumina suspensions to achieve a mirror-like finish, and chemical or electrolytic etching to accentuate microstructural details [5].

In particular, the mounting step provides a stable base for handling small or irregularly shaped samples, which can be difficult to manipulate otherwise. In addition, the mounting material encapsulates the sample, protecting its edges and delicate features during grinding and polishing. Press mounting creates a uniform, flat mount, making it easier to standardize the preparation process for multiple samples. The process can provide better retention of sample edges compared to other mounting methods. In addition, mounted samples are easier to label, store, and transport.

Despite their importance, conventional metallography mounting systems face significant limitations, such as consisting of nearly 100 pieces, complex designs with proprietary components that complicate repairs, leading to prolonged downtime [6–11]. These devices often integrate hydraulic presses with electrically heated devices, which requires specialized maintenance. In addition, the trade war initiated by the United States [12] has shown that Latin American countries face significant technological dependence that drives high costs, such as commercial metallographic mounting systems ($5000 USD to $10,000 USD). Such expenses are prohibitive for institutions in regions with resource constraints, including public schools and universities in Latin America [13]. For example, Mexican technical schools lack access to advanced metallography equipment, stifling hands-on training in materials characterization, a competency critical to industrial workforce development.

Several studies have shown that self-developed solutions can reduce costs while maintaining performance comparable to commercial systems [14,15]. To bridge this gap, this work proposes a novel, compact, low-cost (<$500 USD), and easy-to-manufacture mounting device. The proposed system is designed to produce high-quality polymer encapsulated samples comparable to those obtained from commercial equipment. It is maintenance-efficient and compatible with any commercially available hydraulic press, making it accessible to a wider range of users. In addition, it incorporates an economical and readily available heating system. By democratizing access to metallography technology, this device aims to foster greater interest and training among students, empowering them to explore and contribute to this critical field.

## 2. Materials and Methods

The House of Quality (HoQ), a key tool within the Quality Function Deployment (QFD) methodology [16,17], was used in the design of this device to ensure that it met both customer needs and technical requirements effectively. During the process, customer needs were determined. Then, these needs were translated into specific engineering characteristics using the HoQ matrix, which established clear relationships between customer demands and design parameters. Through this process, they were prioritized, ensuring that the most important aspects were addressed first. The data obtained from the HoQ included a prioritized list of customer requirements, their relative importance, and their correlation with technical specifications, as well as benchmarks comparing the device's performance against competitors. This structured approach not only streamlined the design process but also ensured that the final product was in line with the expectations of the customer. Ultimately, the use of the HoQ in QFD contributed to a more efficient development process, reduced risks, and a device that successfully met market demands.

## 3. Design and Modeling

To ensure that a system capable of withstand the stresses generated by the pressure applied during the operation, Equation (1) was used because only tangential stresses were considered in the analysis [18].

$$\sigma_t = \frac{P \cdot d}{2 \cdot e}, \tag{1}$$

where P is the pressure (Max. 29 MPa based on the pressure recommendations of the polymer supplier), d is the inner diameter (31.75 mm) and e is the minimum wall thickness to use (6.35 mm).

In order to ensure that the device's operation is safe, it was validated using the finite element method (FEM). The modeling considered the following aspects: the program used for the numerical analysis was COMSOL Multiphysics 6.3 [19]. In addition, the main parameters used in the FEM were symmetric 2D geometry [20]; triangular element type;

2387 elements; 439 nodes; and a linear shape function. Although the elastic properties of the polymer may have a significant influence on the device deformation state, the transient effects were not studied. The focus is to conduct a static simulation at the final temperature and pressure where the polymer is already molten, that is, it can be treated as an incompressible liquid within the chamber, and this assumption was used in the analysis applying a pressure of 29 MPa to the walls housing the encapsulated component (blue lines) and the lower face of the base (E) and the upper face of the piston (A) were fixed constraints as an initial boundary condition (red lines) as shown in Figure 1. Due to being a structural contact model, it was used as a form assembly to create contact pairs [21].

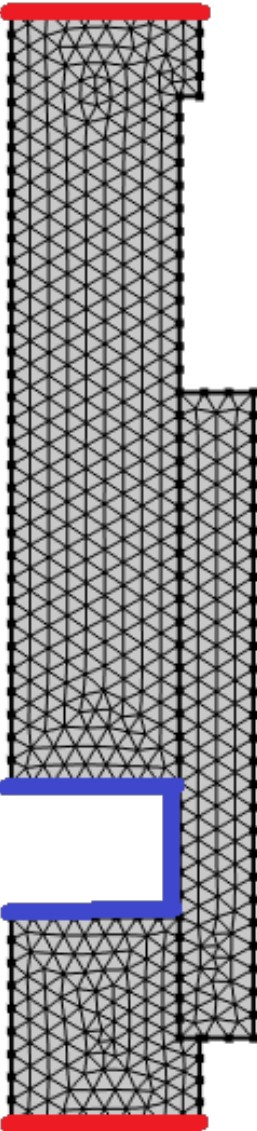

**Figure 1.** Cross-sectional view of the modeled device where red lines denote fixed constraints, while the blue lines show the maximum permissible pressure within the encapsulation material.

Since the device will be subjected to cyclic loads, it is appropriate to determine the fatigue limit or, where applicable, the fatigue safety factor, rather than relying solely on static calculations as shown in Equation (1). In this context, for steels [22,23], the fatigue limit, $\sigma_e$, can be approximated using Equation (2):

$$\sigma_e = 0.504 k_a k_b k_c k_d k_e \sigma_{ul}, \tag{2}$$

where $\sigma_{ul}$ corresponds to the ultimate tensile strength of steel and the k coefficients are the Marin coefficients.

For a non-zero mean stress ($\sigma_m$) can be used the Modified Goodman Criterion, Equation (3):

$$\frac{\sigma_a}{\sigma_e} + \frac{\sigma_m}{\sigma_{ul}} \leq 1 \tag{3}$$

where $\sigma_a$ is the stress amplitude.

Then, to calculate the fatigue safety factor ($n_f$), Equation (4) can be used.

$$n_f = \frac{1}{\frac{\sigma_a}{\sigma_e} + \frac{\sigma_m}{\sigma_{ul}}} \tag{4}$$

## 4. Results and Discussions

Figure 2 shows the House of Quality (HoQ), the core component of Quality Function Deployment (QFD), to design the metallography sample mounting device that bridges the gap between customer needs and technical specifications.

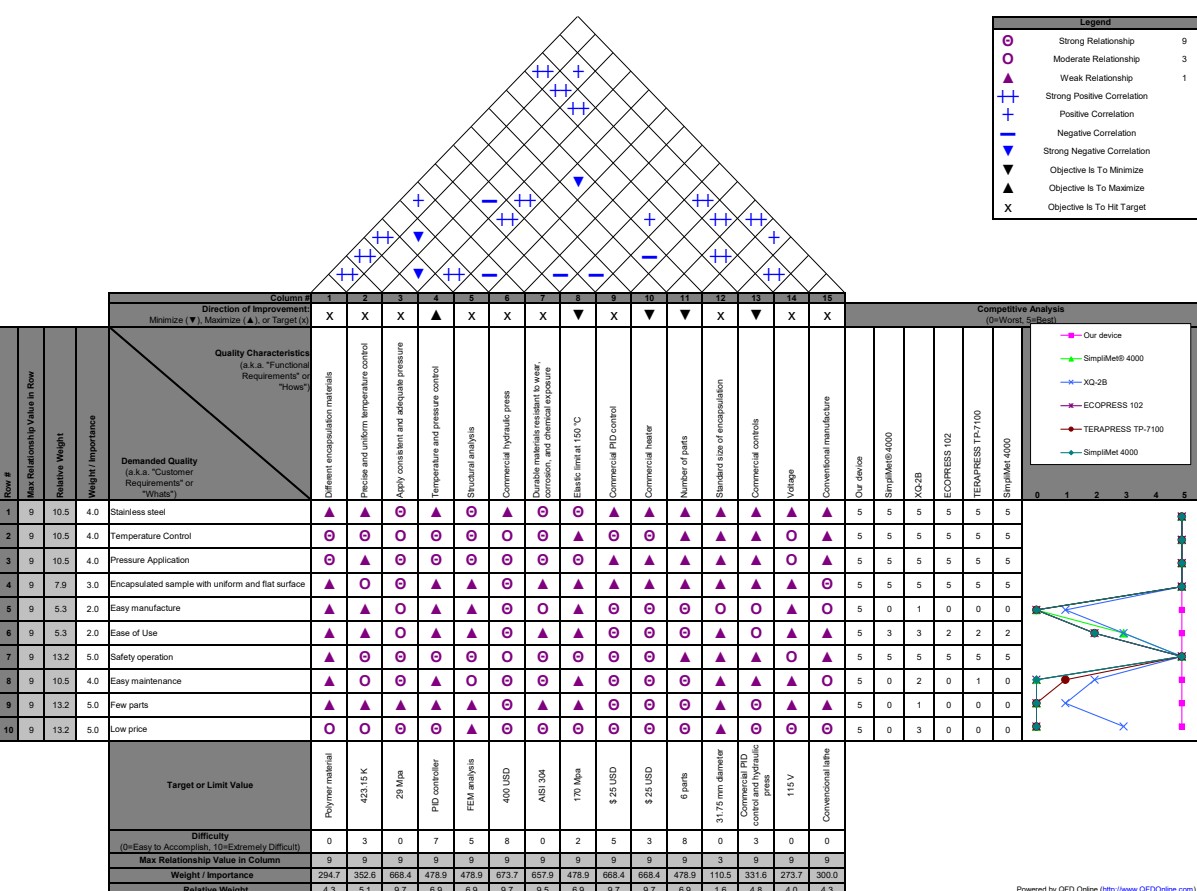

**Figure 2.** House of Quality (HoQ) of Quality Function Deployment (QFD).

The HoQ analysis prioritized customer requirements (e.g., safety, low cost, ease of manufacturing, compatibility with commercial hydraulic presses, and maintenance efficiency) and aligned them with technical specifications between user needs and design parameters. Key outcomes include: material selection (AISI 304 stainless steel) was driven by durability and thermal resistance requirements; dimensional tolerances (for example, sliding H7/g6 tolerance for the piston-ring interface) were optimized for pressure resistance and manufacturability; heating system design (115 V electric band + thermocouple control)

addressed precise temperature management while minimizing costs; benchmarking against competitors highlighted superior cost-effectiveness, adaptability and ease of replication using conventional lathes.

The HoQ framework confirmed that the device successfully balances technical rigor (e.g., FEM-validated stress limits) with accessibility, making it ideal for educational and industrial settings. This structured QFD approach ensured that the final design met priority needs such as safety (through validation of FEM), affordability, and compatibility with existing infrastructure, as reflected in the device's successful performance in producing high-quality metallography samples.

In this sense, the device was designed considering factors such as user requirements as shown in Figure 2: sample geometry, heat and pressure control, user-friendly, few parts, cost, etc. The versatility of enabling an encapsulated sample with a uniform and flat surface was a key consideration. A conductive heating method, capable of melting different thermoplastics, known for its high temperature tolerance, was selected. A 200 W commercial 115 V electric heating band resistance with a K-type termocouple sensor, model GB/T4706.1-2005 from LIXH Inc. (San Gabriel, CA, USA), was chosen as the heat source. A REX-C100 digital thermostat temperature controller and an SSR 40DA solid relay were used. In addition, the device was adapted to a 10 Ton commercial hydraulic press from BGS Technic Mexico Inc. (El Marqués, Mexico), making it compact, economical, and easy to manufacture, operate, and maintain.

Figure 3 shows the assembly of the mounting device of the metallography specimen.

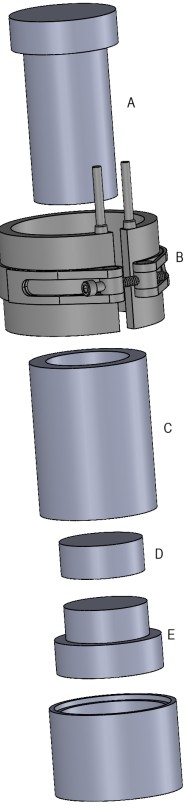

**Figure 3.** Exploded view drawing of the metallography specimen mounting device containing (A) the pressure piston, (B) heater, (C) heating ring, (D) encapsulated sample, (E) base and (F) expelling ring.

The metallography sample mounting device features a cylindrical heating ring (C) with an inner hole for sample placement. The ring is mounted on a base (E) and is compressed by a pressure piston (A). An expelling ring (F) facilitates sample (D) removal. The heating

band (B) wrapped around the ring provides the necessary heat, while a thermocouple and a temperature controller ensure precise temperature control.

It is important to note that the device can be replicated using any conventional lathe. Figure 4 shows the cross-sectional view of the device where the most critical dimensions are the internal diameter of the ring, which can be customized to meet specific requirements, and the wall thickness designed to withstand the pressures recommended by the polymer supplier to be used. In this case, the ring (C) has an inner diameter of 31.75 mm, a wall thickness of 6.35 mm and 50.8 mm long; the piston (A) must have the same inner diameter of the ring with a sliding fit tolerance of H7/g6. The thickness of the sample encapsulation (D) depends on the amount of polymer used. All dimensions can be derived from Figure 4, based on the internal diameter of the ring of 31.75 mm. Although aluminum alloy was initially considered due to its low cost and machinability; however, stainless steel was ultimately preferred, primarily because it offers greater hardness and thus higher resistance to scratching and impacts while remaining more cost-effective than a tool steel. Another issue is that the fatigue limit ($\sigma_e$) of aluminum alloys is typically half that of stainless steel; for reference [24], stainless steels (e.g., 304, 316) often have fatigue limits ($\sigma_e$) of $\approx$200–400 MPa (depending on grade and treatment), while aluminum alloys (e.g., 6061-T6) typically exhibit $\sigma_e \approx$ 100–150 MPa. In this sense, all parts of the device were manufactured from AISI 304 stainless steel.

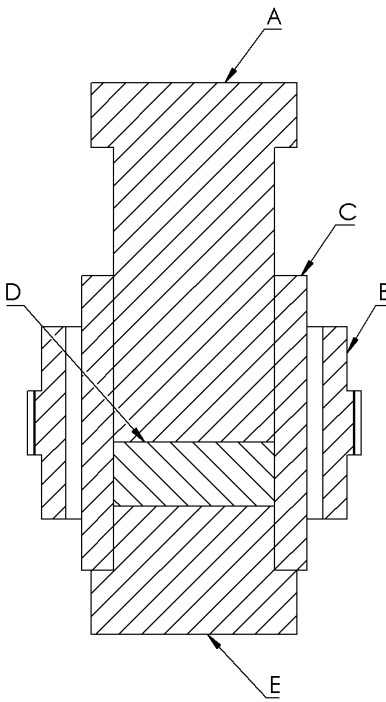

**Figure 4.** Cross-sectional view of the metallography specimen mounting device parts: (A) the pressure piston, (B) heater, (C) heating ring, (D) encapsulated sample, (E) base.

The internal diameter of the ring can be customized to meet specific requirements such as 25.4 mm, 31.75 mm diameter, or any other standard size, and the wall thickness will be designed to withstand the pressures recommended by the supplier of the polymer to be used. In metallography, cylindrical samples are key because many analytical systems, such as grinders and polishers, are designed to handle and process cylindrical mounts. This standardization ensures consistent sample preparation, facilitates automated processes, and allows a precise evaluation of the microstructure of the material.

Although Figure 5 shows that the Von Mises stress values obtained by the finite element method (FEM) have a stress concentration of $\sim$80 MPa, the device perfectly

supports these stresses, as the tensions do not exceed the elastic limit of the material at 140 °C (413.15 K) at 170 MPa) [25].

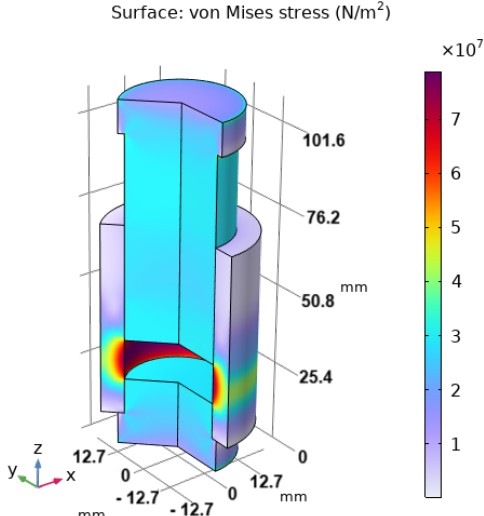

**Figure 5.** von Mises stress distribution under 29 MPa pressure over encapsulated sample.

In this sense, the FEM analysis faced with analytical calculation (Equation (5)) matches the FEM result (von Mises stress = 80 MPa) including a safety margin equal to 2.1 as shown in Equation (6).

$$\sigma_t = \frac{29\,\text{MPa} \cdot 31.75\,\text{mm}}{2 \cdot 6.35\,\text{mm}} = 72.5\,\text{MPa}, \tag{5}$$

$$FoS = \frac{Yield\ Strength\,(170\,\text{MPa})}{Max\ Stress\,(80\,\text{MPa})} = 2.1, \tag{6}$$

Since the device will be subjected to cyclic loads, that is, stress variations that fluctuate between zero and a specific maximum value (similar to what occurs in a gear tooth), both the mean stress ($\sigma_m$) and the stress amplitude ($\sigma_a$) will each be equal to half of this maximum value (~80 MPa). Consequently, they will be equal to each other in this case ($\sigma_m = \sigma_a = 40$ MPa). Assuming $\sigma_e \approx 200$ MPa (Equation (2)) and $\sigma_{ul} \approx 600$ MPa, the fatigue safety factor ($n_f$) can be calculated from Equation (4). The fatigue safety factor ($n_f$) is 3.75, indicating that the steel will withstand infinite cycles under these operating conditions.

In the hand, the assembly procedure for the metallography sample mounting device can be divided into two steps, described below.

Step 1: Sample molding assembly

As shown in Figure 4, the heating ring (C) is mounted on the base (E) and is externally surrounded by the heating band (B), which provides heat. The sample is placed on top of the base (E), and the polymer pellets are poured in. The pressure piston (A) is inserted into the top of the heating ring. Finally, the desired pressure and temperature are applied to obtain the encapsulated specimen (D).

Step 2: Sample demount assembly

To eject the sample, the expulsion ring (F) is attached to the bottom of the assembly from Step 1, as shown in Figure 6. Then, a pressure is applied again. Always, the device must be firmly placed on the press and cooled below 50 °C (323.15 K) before sample retrieval.

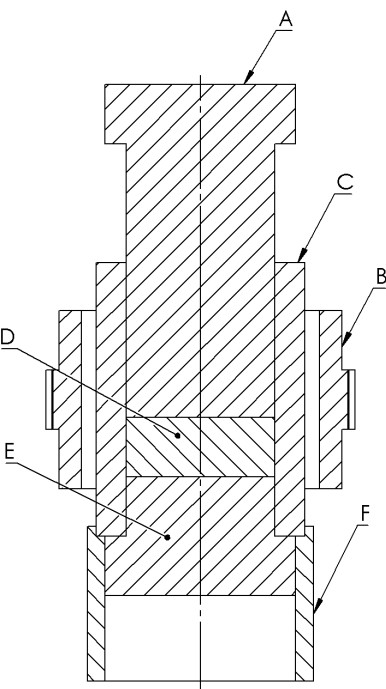

**Figure 6.** Cross-sectional view of the metallography specimen mounting device: (A) the pressure piston, (B) heater, (C) heating ring, (D) encapsulated sample, (E) base and (F) expelling ring.

Figure 7 shows the mounting device for the metallography sample mounted on a commercial hydraulic press and heating system.

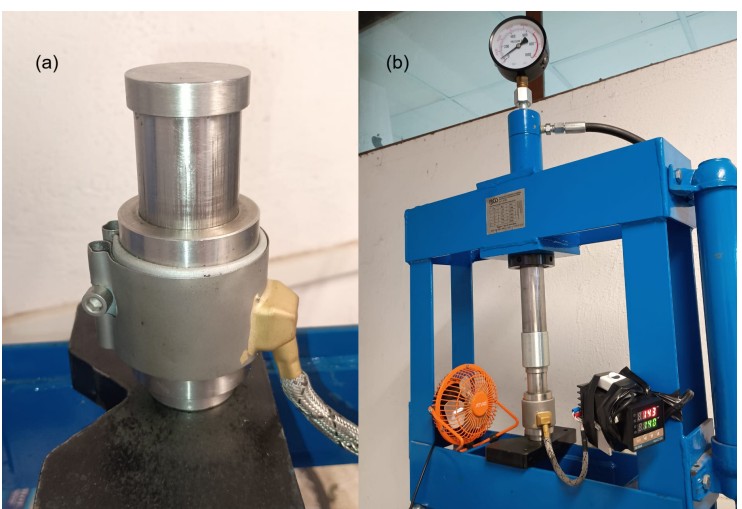

**Figure 7.** (**a**) Metallography specimen mounting device and (**b**) device placed in a hydraulic press.

Using a commercial PhenoCure powder from Buehler Inc. (item No. 203300400, Lake Bluff, IL, USA), Figure 8 shows the encapsulated sample obtained from the device applying a pressure of 29 MPa, at 140 °C (413.15 K) and a dwell time of 5 min in its stable state, showing that the sample was obtained with the same quality as that using commercial devices [26]. For reference, Struers Inc. (Westlake, OH, USA) claims that their CitoPress models can achieve minimum heating and cooling times of 1 min each. In comparison, the designed device reaches the desired temperature in just over 2 min and cools to 50 °C in approximately 4 min with the help of a small fan, as shown in Figure 7. This allows to conclude that the times achieved by our device are acceptable.

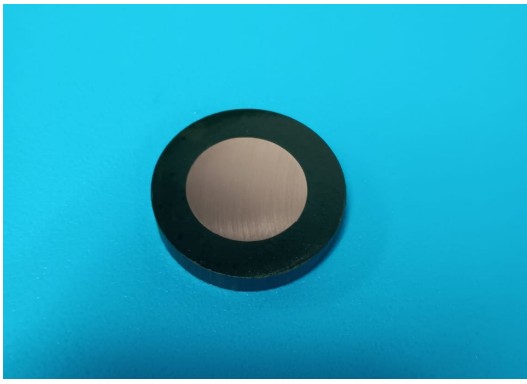

**Figure 8.** Sample obtained from the metallography specimen mounting device.

Table 1 shows the benchmarking of key parameters against commercial systems.

**Table 1.** Comparison between proposed device and commercial systems.

| Parameter | Proposed Device | Commercial Systems |
|---|---|---|
| Cost | <$500 USD | $5000–$10,000 USD |
| Component Count | 6 | ~100 |
| Safety Factor (FoS) | 2.1 | 1.5–2.0 (industry standard) |
| Heating System | Commercial | Proprietary, integrated |
| Compatibility | Standard hydraulic press | Specialized equipment |
| Heat up and cooling time | ≥6 min | ≥2 min |

Table 1 displays a summary of the main features that compare the proposed device with other commercial systems, from which it can be clearly concluded that although the device requires slightly longer to obtain a sample, it adequately meets the established requirements. The design of the device allows for mounting a metallography sample, allowing for easy handling and conditioning of the surface of interest.

## 5. Conclusions

This study successfully addresses the critical challenge of high cost and complexity in the preparation of metallography samples by introducing a novel mounting device that combines affordability, durability, and performance. Key achievements include: cost reduction at <$500 USD vs. commercial prices of $5000–$10,000 USD, making it accessible to educational institutions and industries in resource-constrained regions; with only 6 components (vs. 100 in conventional systems), the device ensures ease of manufacturing, repair, and replication using standard lathes; validated via FEM, the device operates safely under 29 MPa pressure with a static safety factor of 2.1 matching commercial standards in edge retention and surface quality; the structured QFD/HoQ approach ensured alignment of customer needs (e.g., safety, compatibility) with technical specifications, while benchmarking highlighted superior adaptability and cost-effectiveness; by enabling high-quality sample preparation with minimal infrastructure, this device democratizes metallography training and supports workforce development in regions like Latin America, where technological dependence and budget constraints hinder progress. This innovation not only bridges a critical technological gap, but also empowers global scientific and industrial communities to advance materials research sustainably.

## 6. Patents

Utility Model Title No. 5628. https://vidoc.impi.gob.mx/visor?d=MX/2024/108782 (accessed on 1 April 2025).

**Funding:** This research received no external funding.

**Institutional Review Board Statement:** Not applicable.

**Informed Consent Statement:** Not applicable.

**Data Availability Statement:** The original contributions presented in this study are included in the article. Further inquiries can be directed to the corresponding author.

**Acknowledgments:** We express our gratitude for the support provided by technician Fabian Perez-Cornejo.

**Conflicts of Interest:** The author declares no conflicts of interest.

## Abbreviations

The following abbreviations are used in this manuscript:

| | |
|---|---|
| HoQ | House of Quality |
| QFD | Quality Function Deployment |
| FEM | Finite Element Method |
| FoS | Factor of Security |

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
