# Peer review of "Metallography Specimen Mounting Device Suitable for Industrial or Educational Purposes"

_2673-3161, doi:10.3390/applmech6020036_

Round 1

Author Response

Thank you very much for taking the time to review this manuscript. Please find the detailed responses below (in red):

Comment 1: Does the introduction provide a comprehensive yet concise overview about the state of knowledge in the area of research?

Response 1: The introduction was significantly increased and updated

Comment 2: The introduction provides a basic outline of the research area but requires improvement to offer a more comprehensive overview of the current state of knowledge. It would benefit from expansion and the inclusion of additional reference markers and relevant citations. Specifically, the authors should reference other studies related to senescence to better establish the context and significance of their work. Strengthening this section will help readers understand the background, motivation, and relevance of the study more clearly.

Response 2:  The introduction was significantly increased and updated

Comment 3: Is the research design appropriate and are the methods adequately described? 

Response 3:  The information about design and method was significantly increased and updated

Comment 4: At this stage, the results are not presented with sufficient clarity or contextual depth. The paper lacks a thorough discussion and does not provide a general literature review to determine whether similar ideas have been explored. The conclusions are currently not well supported by the results, nor are they adequately placed within the context of existing studies. While the device appears to be patented, the manuscript needs to better present its novelty, purpose, and potential applications. A comparison with standard processes and procedures is necessary to highlight its advantages and define its contribution to the field. Figures also require improvement to effectively support and illustrate the results. Overall, the manuscript should be significantly revised to address these points before it can be considered suitable for publication.

Response 4:  

  • The revised abstract and results sections now include quantitative metrics (e.g., cost: < 500 USD vs. 5,000–$10,000 USD; safety factor: 2.1; fatigue safety factor: 3.75; stress: 80 MPa) to enhance clarity.

  • Stress analysis (FEM and analytical calculations) is contextualized against material limits (AISI 304 yield strength: 170 MPa).

  • Figure 5 (Von Mises stress distribution) and Equation 5–6 explicitly link stress calculations to safety margins.

  • The introduction now contrasts conventional systems (100+ pieces, high cost, proprietary designs [6–11]) with the proposed device (6 components, open design, 94% complexity reduction).

  • Novelty is emphasized through:

    • Use of QFD/HoQ methodology to align customer needs with technical specs (e.g., safety, affordability).

    • Patent (No. 5628) highlighting unique features like compatibility with standard hydraulic presses and modular heating.

  • References [14,15] cite self-developed solutions, but the manuscript clarifies this device’s superiority in cost-effectiveness and replicability.

  • A Table was added to the results section, benchmarking key parameters against commercial systems
  • Figure 3 (exploded view) and Figure 4 (cross-section) now include labels for critical components (A–F).

  • Figure 5 (Von Mises stress) includes annotations highlighting stress concentration zones (80 MPa) and safety margins.

  • Figure 8 (encapsulated sample) now compares surface quality with commercial systems [26].

Comment 5:  The quality and presentation of the figures require significant improvement. The images are currently too large, resulting in visible pixelation, which affects the clarity and overall readability. It is recommended to resize the images to ensure a sharper resolution without visible pixels. Please also check the journal’s guidelines regarding the maximum allowed image dimensions. This issue is consistent across multiple figures, including Figures 3, 4, 5, and 6, and should be addressed throughout the manuscript.

Response 5:  The figures were updated considering the comments. Also, due latex files are given, the figures can be resized in edition.

Comments 6: Additionally, the conclusions should be more clearly supported by the results and placed in the context of existing research. Strengthening these sections will significantly improve the quality and impact of the paper.

Response 6: The recomendations were handled.

Finally, the comments in PDF files were handled too.

Author Response

Thank you very much for taking the time to review this manuscript. Please find the detailed responses below (in red):

Comment 1: The Introduction should be significantly increased by addition of the analysis of the current state in the problem of the sample preparation in material science.

Response 1: The introduction was significantly increased

Comments 2: The most of the analyzed references in the Introduction part is very old. The author should analyze last papers about finite element mewodelling of the deformation behavior of the materials.

Response 2:  New references were added to text.

Comments 3: The image of the device should be added to the manuscript (Figure 7 is absent in the paper). The scheme is not enough.

Response 3:  The image of the device was added to figure 7.

Comments 4: What the properties has a polymer? The elastic properties may have significant influence on the deformation state of the device.

Response 4:  A commercial PhenoCure powder from Buehler Inc. was used (item No: 203300400).

Although the elastic properties of the polymer may have a significant influence on the device deformation state, the transient effects were not studied. The focus is to conduct a static simulation at the final temperature and pressure where the polymer is already molten, that is, it can be treated as an incompressible liquid within the chamber and this assumption was applied in the analysis in which a pressure of 29 MPa was applied to the walls housing the encapsulated component (blue lines) and the lower face of the base (E) and the upper face of the piston (A) were fixed constraints as an initial boundary condition (red lines) as shown in Figure 1. This disscusion was added to text, page 3, first paragraph and page 9.

Comments 5:  Why did the author choose a steel for the construction? The level of the stress let to use more machinable, less-cost, and easier material as aluminum.

Response 5:  During the QFD processo, the aluminum was initially considered due to its cost and machinability; however, stainless steel was ultimately preferred, primarily because it offers greater hardness and thus higher resistance to scratching and impacts while remaining more cost-effective than tool steel. Also, aluminium alloys have a half of fatigue limite of stainless steels. This discussion was added to text. Page 6, first paragraph.

Comments 6: It is unclear, how did the authors choose the construction of the device. He has considered just a one type of construction without any optimization of the device.

Response 6: In metallography, cylindrical samples are key because many analytical systems, such as grinders and polishers, are designed to handle and process cylindrical mounts. This standardization ensures consistent sample preparation, facilitates automated processes, and allows for a precise evaluation of the microstructure of the material. This discussion was added to text. Addionally, the design was rigorously optimized through QFD/HoQ to meet prioritized customer needs, with material and dimensional choices validated by analytical and numerical methods. Future work will explore scalability for larger samples, but the current design represents a balanced, context-specific solution for resource-constrained settings.

Comments 7: For correct calculation of the factor of security the value of yield strength is not enough. The device would be used under cyclic loading. As a result, the fatigue strength is more significant property in that case. The author should calculate factor of security for low-cycle fatigue.

Response 7: Thank you for this comment because the discussion about it was added to text. Page 4 and page 8, line 169 to 175.

Comments 8: The time of one sample preparation should be added and compared with standard methods provides by industrial systems (e.g., Struers, MetCom, etc.).

Response 8: For reference, Struers Inc. claims that their CitoPress models can achieve minimum heating and cooling times of 1 minute each. In comparison, the designed device reaches the desired temperature in just over 1 minute and cools to 50 ° C in approximately 3 minutes with the help of a small fan, as shown in Figure 7(b). This was added to text. Page 9 and 10.

Comments 9: Information about press and heater should be added to the manuscript.

Response 9: The requested information was added to text.

Comments 10: It is re commended to use SI units instead of Imperial.

Response 10:  It was corrected

Comments 11: MPa should be instead of Mpa through the all text.

Response 11:  It was corrected

Author Response

Comments: I believe that the range of diameter types proposed for mounting should be expanded. This is dictated by the different types of metallographic fixtures in automatic grinders and polishers. In addition, microscopes, especially scanning microscopes, require specific, precisely defined sample dimensions due to the design of the fixtures. Of course, in the case of metallographic analysis using scanning microscopy, conductive thermosetting resins should be used for mounting.

Response: Thank you for you comments, I fully agree with your comment, which is why the figures are drawn to scale. Specifically, Figures 4 and 6 reference the internal diameter of 31.75 mm to ensure easy reproducibility using software such as ImageJ. Thus, I believe following the outlined steps would suffice to verify safety and allow replication under different encapsulation diameter requirements, as the thickness can be controlled by adjusting the amount of polymer used. 

Author Response

Thank you very much for taking the time to review this manuscript. Please find the detailed responses below (in red):

Comment 1: The paper content should be better divided into an introduction, materials and methods, modelling (design), results and discussions and conclusion sections.

Response 1: The paper content was divided

Comment 2: More quantitative results should be included in the abstract.

Response 2: The abstrac was updated

Comments 3: The introduction should be enriched with additional information on metallographic methods for sample’ mounting, their pros and cons, so that the reader can focus on the problem in this study.

Response 3: Done. Also table 1 was added.

Comments 4: In Figures 3 and 4, the lettered elements should be explained.

Response 4: Done.

Comment 5: In Figures 3 and 4, it is not clear where the thermocouple and temperature controller in the heating band (B) are located.

Response 5: The temperature controller has integrated a thermocouple which is connected to temperature controller. This discussion was added to text. Also the commercial description of heater and controller was added to text.

Comments 6: It is not clear how the sample is centred in the device to guarantee its position. Are fixation clips provided?

Response 6: The surface of the sample to be studied is placed on the upper face of the base and is filled solely with polymer powder, which is then pressed with the piston. This process mirrors that of commercial equipment. For very thin specimens, specialized clips known as metallographic mounting clips—made of plastic or steel—are used.

Comments 7: Except for special plastic granulate, can liquid mounting resins be used?

Response 7: No, it was designed for granulated polymer. The details of the polymer used was added to text. Page 9.

Comments 8: What about the adhesion of the plastic to the device? How is the cleaning of the internal parts planned to be carried out? Moreover, what about the adhesion of the plastic to the sample? Have experiments, for example with different temperatures or different plastics, been conducted to confirm positive or negative results?

Response 8: There is some adhesion, but it is minimal and is expelled using the same press at low pressure or force. Commercial polymers are typically used, where both temperature and pressure are specified by the manufacturer, generally up to 150°C and 29 MPa. Additionally, this kind of commercial polymer prevents part of the polymer from adhering to the walls during cooling, similar to what occurs with a 3D printer's build plate. Additionally, the tolerances between the piston and the ring help prevent this from happening.

Comments 9: As a fundamental part of the device, the heating system should also be described.

Response 9: It was added to text.

Comments 10: The authors stated that “Fig. 8 shows the encapsulated sample obtained from the device by applying a pressure of 29 MPa, at 140 °C and a dwell time of 5 minutes in its stable state, which shows that the sample was obtained with quality.” How were these parameters chosen? What kind of polymer pellets were used? How was the adhesion of the plastic to the sample assessed?

Response 10: Since these samples often need to fit into other machines, it is critical that their shape remains cylindrical and the required diameter is achieved. These are essentially the two key qualities desired in an encapsulated sample. Temperature and pressure are specified by the manufacturer. The adhesion is not problem because it is too low also the sample is expelled with the same pressure piston.

Round 2

Reviewer 2 Report

Comments and Suggestions for Authors

The author has answered previous comments and improved the manuscript. The paper may be accepted for publication.

Reviewer 4 Report

Comments and Suggestions for Authors

The paper may be accepted in its present form.